# Reproducibility and Reliability of Spectralis II OCT Angiography Vascular Measurements

**DOI:** 10.3390/diagnostics12081908

**Published:** 2022-08-07

**Authors:** Marco R. Pastore, Alberto Grotto, Francesca Vezzoni, Andrea Gaggino, Serena Milan, Stefano Gouigoux, Pier Luigi Guerin, Alex L. Vinciguerra, Gabriella Cirigliano, Daniele Tognetto

**Affiliations:** Eye Clinic, Department of Medicine, Surgery and Health Sciences, University of Trieste, 34129 Trieste, Italy; dr.albertogrotto@gmail.com (A.G.); vezzoni.francesca@gmail.com (F.V.); andrea.gaggio@gmail.com (A.G.); serena.milan2@gmail.com (S.M.); s.gouigoux@gmail.com (S.G.); pierluigi.guerin@gmail.com (P.L.G.); alexluciavinciguerra@libero.it (A.L.V.); gabriellacirigliano16@gmail.com (G.C.); tognetto@units.it (D.T.)

**Keywords:** retinal microvasculature, optical coherence tomography angiography, reliability, reproducibility, foveal avascular zone, vessel density

## Abstract

Purpose: to investigate the reproducibility and reliability of OCT-A vascular measurements using Heidelberg Spectralis II OCT-A. Methods: a prospective study involving a single eye of patients aged 18 or older with no ocular disease. In order to investigate the reliability of the first and second OCT-A scans, the coefficient of variation of the foveal avascular zone (FAZ) and the vessel density (VD) in the superficial (SCP), intermediate (ICP) and deep capillary plexus (DCP) were calculated. Results: A total of 75 eyes were included in the study. The mean FAZ in the first and second scan was 0.36 × 0.13 mm^2^ and 0.37 × 0.12 mm^2^, respectively, in the SCP, 0.23 × 0.10 mm^2^ and 0.23 × 0.09 mm^2^ in the ICP, and 0.42 × 0.11 mm^2^ and 0.43 × 0.12 mm^2^ in the DCP. The overall VD was 36.05 × 9.01 and 35.33 × 9.92 at the first and second scan, respectively, in the SCP, 21.87 × 5.00 and 21.32 × 5.56 in the ICP, and 23.84 × 6.53 and 23.20 × 6.83 in the DCP. No statistically significant differences in FAZ measurements and VD in all sectors of each capillary plexus were observed between the first and second scan (*p* > 0.05). Conclusion: our study demonstrated the good reproducibility and reliability of OCT-A vascular measurements in the analysis of the FAZ and the quantification of VD in each capillary plexus of the retina.

## 1. Introduction

Optical coherence tomography angiography (OCT-A) is a newly introduced imaging technique that allows the visualization of functional blood vessels in the eye. The principle of OCT-A relies on the variation in the optical coherence tomography signal caused by moving particles, such as red blood cells, through a motion contrast algorithm [1].

Since OCT-A has been introduced in clinical practice, microvascular retinal architecture can be tridimensionally studied and different vascular layers clearly analyzed.

Retinal circulation consists of four main plexuses: the radial peripapillary capillary plexus, superficial capillary plexus (SCP), intermediate capillary plexus (ICP) and deep capillary plexus (DCP) [2,3,4,5].

The foveal avascular zone (FAZ) is the very central region of the fovea, so called because it lacks blood vessels, which ends delineating a circular-shaped area, whose dimensions differ greatly in individuals, even in the normal range [6].

Fluorescein angiography (FA) has been the gold standard for retinal vasculature imaging since the 1960s. It is an invasive technique that requires a dye injection to investigate the vascularization of the retina. Nevertheless, it does not allow the visualization of the deeper macular capillary plexuses and it does not evaluate the blood flow and the vessel density (VD) [6,7]. 

Conversely, OCT-A requires no dye injection and can capture images of the different vascular layers. In addition, it gives the opportunity to directly compare images with previous acquisitions in follow up records.

Many different OCT-A devices are available nowadays with an own-producer-related algorithm to generate vascular images for clinical routine activity and research issues [8]. Moreover, different software and algorithms are used for OCT-A image analysis [9]. As a result, different values can be observed when not using the same OCT-A device and software to investigate retinal vascularization parameters. 

Important data assessed with OCT-A analysis includes the VD quantification, which is extremely useful to estimate retinal perfusion state [10]. To the best of our knowledge, only two reports analyzed the reliability of macular microvasculature measurements by Heidelberg Spectralis II OCT-A, but using its own specially developed software in a restricted population composed of subjects with no ocular diseases [8,11].

Our study aimed to investigate the reproducibility and reliability of OCT-A vascular measurement in a large cohort study of healthy eyes using Heidelberg Spectralis II OCT-A.

## 2. Methods

A prospective study was conducted at the University Eye Clinic of Trieste between October and December 2020. It was performed in conformity with the Declaration of Helsinki and the standards of Good Clinical Practice. The protocol was approved by the Institutional Review Board and written informed consent was obtained from all participants.

A single eye of patients who were at least 18 years old with best-corrected visual acuity (BCVA) of 20/20 and no clinical evidence of any ocular disease were included in the analysis. Exclusion criteria were refractive error ≥ 6 diopters and a history of previous laser treatment or ocular surgery. Eyes with poor-quality OCT-A images (signal strength index (SSI) lower than 25) due to eye media opacities or eye movements during the examination were also excluded.

All subjects underwent a complete ophthalmologic examination, including assessment of BCVA measured at 4 m with standard Early Treatment for Diabetic Retinopathy Study (ETDRS) charts, Goldmann applanation tonometry, slit lamp biomicroscopic examination and fundus examination.

OCT-A images were assessed using Heildelberg Spectralis II OCT (Software Version 6.15, Heidelberg Engineering, Heildelberg Germany). En face images of both eyes were acquired in High-Resolution mode with a 10° × 10° angle and a lateral resolution of 6 μm/pixel, resulting in a retinal section of 2.8 mm × 2.8 mm for the visualization of the capillaries plexus and an axial resolution of 3.9 microns per pixel yielding precise multilayer segmentation.

At the end of acquisition, only one eye of a single subject was randomly chosen and included in the analysis. A second OCT-A scan for the studied eyes was performed by the same investigator within one day. Before the analysis, all included scans were revised for artefacts or shadows. The measurements of VD and FAZ were performed and analyzed by two different experienced examiners (F.V. and A.G.) and compared according to the Cohen κ coefficient. The first and second VD and FAZ measurement of a single eye were obtained by the same physician. 

The region of interest (ROI), which in our study corresponded to the whole area of analysis, centered on the fovea, measured 2.8 mm × 2.8 mm. ROI images were split into 9 sectors of the same size using an online software called pinetools.com, as in Figure 1. Each sector of SCP, ICP and DCP was then uploaded and processed with the Image J software (S.M., S.G. and P.L.G.). A comparison of ROI images before and after Image J processing is reported in Figure 1. Image J software performed multiple segmentations, allowing analysis of vessel density. The percentage of the sample area occupied by vessel lumens following binary reconstruction was defined as vessel density. The volume assessment process is made up of 3 steps. The first step calculates the pixel number per unit area (1 cm^2^). The second step measures the area of the ROI using the ratio of pixel number per unit area. The final step calculates the volume using the method of the Integral. VD is calculated by creating a binary image of the vessels from the grayscale OCT-A en face image: each vessel pixel is white while each tissue pixel is black. Afterwards, we manually calculated the FAZs at SCP, ICP and DCP. These areas were selected and demarcated manually by interconnecting the most inward projecting vessel’s ends. In order to investigate the reliability of the first and second OCT-A scans, the coefficient of variation of SCP, ICP, DCP and of FAZ area were calculated.

## 3. Statistical Analysis 

Quantitative variables (FAZ and VD measurements for each vascular layer) were expressed as mean and standard deviation (SD), maximum value and minimum value. Scatter plots and Pearson index were used to assess concordance between measurements. Bland–Altman plots and measures of the coefficient of repeatability (CR) were used to evaluate the inter-examiner agreement of the measurements. T-tests were used to evaluate the differences between the two measures for each variable. The coefficients of variation (CV) were calculated for each variable measured by the first examiner to compare the dispersion of the measures of the VD and FAZ area. A *p* value < 0.05 was considered statistically significant. To quantify the intergrader agreement for qualitative variables, the Cohen κ coefficient was computed. Statistical analyses were performed using SPSS software 11.0 (SPSS Inc., Chicago, IL, USA).

## 4. Results

### 4.1. Demographics Results

A total of 75 eyes (38 right eyes and 37 left eyes) of 75 Caucasian subjects (34 males and 41 females) were included in this study. The mean age of the study subjects was 48.75 years (SD, 6.03 years) with a range of 37–58 years. All subjects were in good health with no systemic or ocular disease, presenting a best-corrected visual acuity of 20/20 with a mean refractive error of −0.75 spherical equivalent and intraocular pressure within the normal range. At cross-sectional OCT scan, all eyes had normal foveal morphology.

### 4.2. Foveal Avascular Zone Analysis

The results for the FAZ analysis in the first and the second OCT-A run are reported in Table 1. The mean FAZ in the first and second scan was 0.36 ± 0.13 mm^2^ and 0.37 ± 0.12 mm^2^, respectively, in SCP, 0.23 ± 0.10 mm^2^ and 0.23 ± 0.09 mm^2^ in ICP, and 0.42 ± 0.11 mm^2^ and 0.43 ± 0.12 mm^2^ in DCP. No statistically significant differences in FAZ for SCP, ICP and DCP were observed between the first and second measurement (*p* > 0.05). The CR of the FAZ for each capillary plexus is reported in Table 1. No statistically significant interexaminer differences were observed for the FAZ analysis in the SCP, ICP and DCP, respectively (*p* > 0.05). The CVs between the first and second scan in the FAZ analysis are reported in Table 1, with no statistically significant differences between the two runs (*p* > 0.05). The Bland–Altman plots of the SCP, ICP and DCP demonstrate the good consistency of the first and second scan (Figure 2).

### 4.3. Vessel Density Analysis

In the SCP analysis, the overall VD was 36.05 ± 9.01 and 35.33 ± 9.92 at scan 1 and 2, respectively. The mean VD ranged between 27.02 and 45.48 in the first scan, and between 26.61 and 45.22 in the second (Table 2). In both scans, the minimum value was detected in the inferotemporal sector and the maximum in the nasal one. No statistical differences in VD in the different sectors were observed between the first and second measurements (*p* > 0.05).

In the ICP, a lower overall VD was found, with 21.87 ± 5.00 and 21.32 ± 5.56 in the first and second scan, respectively. The mean VD in each different sector was lower than in the SCP, ranging from 18.16 to 25.55 in the first scan, and from 17.96 to 25.33 in the second analysis (Table 2). No statistical differences in VD for the different sectors in ICP were observed between the first and second measurements (*p* > 0.05).

The overall VD of DCP was 23.84 ± 6.53 and 23.20 ± 6.84 at measure 1 and 2, respectively. The mean VD in DCP ranged between 20.32 and 29.71 in the first scan, and between 19.80 and 29.14 in the second run (Table 2). No statistical differences in VD for the different sectors in DCP were observed between the first and second measurement (*p* > 0.05).

The mean VD in the different sectors in the superficial, intermediate and deep capillary plexuses are reported in Table 3, Table 4 and Table 5, respectively. In particular, the highest value of VD was found in the nasal sector of all plexuses, whereas the lower value of VD was detected in the inferotemporal sector in the SCP and DCP analysis, and in the superotemporal sector in the ICP.

The CR of the VD for each sector is reported in Table 3 for the SCP, in Table 4 for the ICP and in Table 5 for the DCP. No statistically significant interexaminer differences were observed for the VD analysis in the SCP, ICP and DCP, respectively (*p* > 0.05).

The CVs showed no statistically significant difference in the comparison between the first and second run for all capillary plexuses (*p* > 0.05; Table 3, Table 4 and Table 5).

The good reliability of the two scans of each sector in the SCP, ICP and DCP are represented in the Bland–Altman plots (Figure 3, Figure 4 and Figure 5).

## 5. Discussion

OCT-A is a non-invasive imaging technique that has revolutionized ophthalmologic everyday clinical practice, becoming an essential imaging device for the morphological analysis of the retina and for research issues [12].

OCT-A Spectralis II OCT function is based on a motion contrast algorithm between two repeated scans of the same retinal region and allows the visualization of retinal vascular plexuses without dye injection.

Nowadays, several different OCT-A tools are available and each producer employs its own algorithm to generate vascular images based on OCT signal information [9]. Thus, it is important to investigate the corresponding reliability for each OCT-A device and analysis software. Heidelberg Spectralis II OCT uses a probabilistic full-spectrum amplitude decorrelation algorithm (FSADA), which allows an axial resolution higher than the split spectrum decorrelation algorithm, without projection artifacts in retinal layers [13].

Since OCT-A imaging has been demonstrated to correlate with the histological pattern of the chorioretinal vasculature [14], several studies have employed this device to demonstrate many relevant clinical findings including areas of macular telangiectasia, impaired perfusion, capillary remodeling and neovascularization [15,16,17,18,19]. However, it appears to be of crucial importance to verify the reproducibility of the images acquired using this novel device to appreciate the fine evolution of retinal vascular pathologies in the clinical follow up. Therefore, it sounds reasonable to have an insight on vascular flow characteristics in healthy subjects to fully comprehend the pathologic features of the FAZ area and of the VD in all different vascular complexes.

In our study, the images obtained with Heidelberg Spectralis II OCT-A underwent further processing with the Image J software developed by Wayne from the Ocular Oncology Service [20,21]. This step offered a higher contrast definition of the vascular paths in each capillary plexus of the retina, allowing a detailed analysis of the ROI. No significant differences in CV for the FAZ area and vessel density in the SCP, ICP and DCP segmentation analysis were reported, proving the good consistency of image acquisition.

In literature, only two previous reports assessed the reliability and reproducibility of Heidelberg Spectralis II OCT-A analysis for FAZ and SCP or DCP vessel density [8,11]. Lupidi et al., using a Spectralis OCT-A prototype and custom-built software (AngioOCToll), reported a mean vessel density of 27.84 ± 1.93 for the SCP and 27.69 ± 2.23 for the DCP. The FAZ areas of the two layers were 0.28 ± 0.11 mm^2^ and 0.30 ± 0.10 mm^2^ for SCP and DCP, respectively. No data for the FAZ and vessel density in ICP were provided [11]. Compared to our analysis, the different results detected might be related to the different algorithms’ analysis and software machines used, such as the projection artifact removal tool that removes artifacts due to blood movement. Furthermore, Image J software with manual tracks used in our study creates binary images for better contrast and the identification of erroneous autosegmentation.

More recently, Hosari et al. [8] investigated macular microvasculature measurements in 23 eyes of healthy subjects. They performed en face OCT-A scans using Heidelberg Spectralis II OCT and analyzed the vessel density of the different retinal layers in combination with Matlab software, an EA-Tool OCT-A application. The FAZs of the SCP, ICP and DCP were then manually calculated. The authors reported a mean FAZ area of 0.43 ± 0.16 mm^2^ in the SCP, 0.28 ± 0.1 mm^2^ in the ICP, and 0.44 ± 0.12 mm^2^ in DCP. The SCP showed a VD ranging from 30.4 and 33.5 in the first scan and between 30.2 and 33.1 in the second scan. The ICP ranged from 20.9 to 24.7 in the first scan and from 21.2 to 24.9 in the second scan. The DCP’s VD ranged from 23.5 and 27.6 in the first scan and between 23.6 and 27.6 in the second scan. Due to the different mean age of the enrolled subjects corresponding to 48.75 years in our analysis (range, 37–58 years) and 26.8 years in Hosari et al.’s report (range, 19–48 years), a direct comparison between the two studies cannot be fully performed. In addition, even the use of different analysis software is a fundamental bias for the comparative analysis. Furthermore, the center of the macular region, defined as ROI and marked by the user, was larger in our study (7.84 mm^2^) than in the studies by Lupidi et al. [11] (2.86 mm^2^) and Hosari et al. [8] (6.10 mm^2^). This can provide additional explanation of differences among results obtained in the studies.

Our results, as already described by other papers, confirm a larger FAZ at the level of the DCP compared to the SCP [11,21,22]. A possible explanation to understand this finding, as already proposed by Lupidi et al. [20], is that SCP presents a continuous ring of capillaries delineating the avascular zone, whereas the DCP has terminal capillaries without interconnections.

Again, our results appear to be in line with those of Hosari et al. [8], confirming higher overall VD in SCP compared to ICP and DCP. Lupidi et al. [11] previously reported no statistically significant differences in VD between SCP and DCP, indeed not analyzing the ICP. Quite the opposite, Gadde SG et al. [22], in a group of 52 healthy subjects who underwent SSADA OCT-Angiography, found a higher VD in DCP compared to the SCP. This result could be explained, as proposed by Lupidi et al. [11], by the fact that the split spectrum amplitude decorrelation algorithm causes projection artifacts from the more superficial to the deeper retinal layers, so that the SCP would be, at least partially, duplicated on DCP. Heildelberg Spectralis II OCT, used in our study, offers high lateral and axial resolution, granting more precise visualization of the different capillary plexuses and probably could better analyze deeper layers.

Our study had some limitations. We enrolled only Caucasian subjects and no data across other races are available in this report. In addition, patients’ data on retinal microcirculation divided by age-related subgroups were not collected. It is well known that VD and FAZ area directly correlates with age and sex as it was investigated by Coscas et al. [7], who provided age-related VD OCT-A mapping data on three groups of healthy subjects ranging from 20 to 79 years by using the AngioVue OCT-A tool combined with AngioAnalytics software. They found the mean FAZ area to be smaller among people of 60 years or older at the level of the SCP. A transversal age-based study with Heildelberg Spectralis II OCT is mandatory to investigate the role of age in the modification of the central avascular zone size and of the vessel density of the different retinal capillary plexuses. Furthermore, we have to underline that in clinical practice, images obtained with OCT-A are not processed using Image J. The good reproducibility and reliability of OCT-A vascular measurements obtained in our analysis are based on post-processing steps performed by this tool.

## 6. Conclusions

In conclusion, this study highlights the good reproducibility and reliability of OCT-A vascular measurement with Heildelberg Spectralis II OCT-A in healthy subjects. The introduction of VD evaluation appears to be a crucial tool to estimate the retinal perfusion state in routine clinical practice. In our analysis, a good consistency and reliability of Heildelberg Spectralis II OCT-A retinal vascular assessment has been proven, confirming its usefulness in clinical use or research purposes to distinguish between macular vascular alterations and healthy structures.

## Figures and Tables

**Figure 1 diagnostics-12-01908-f001:**
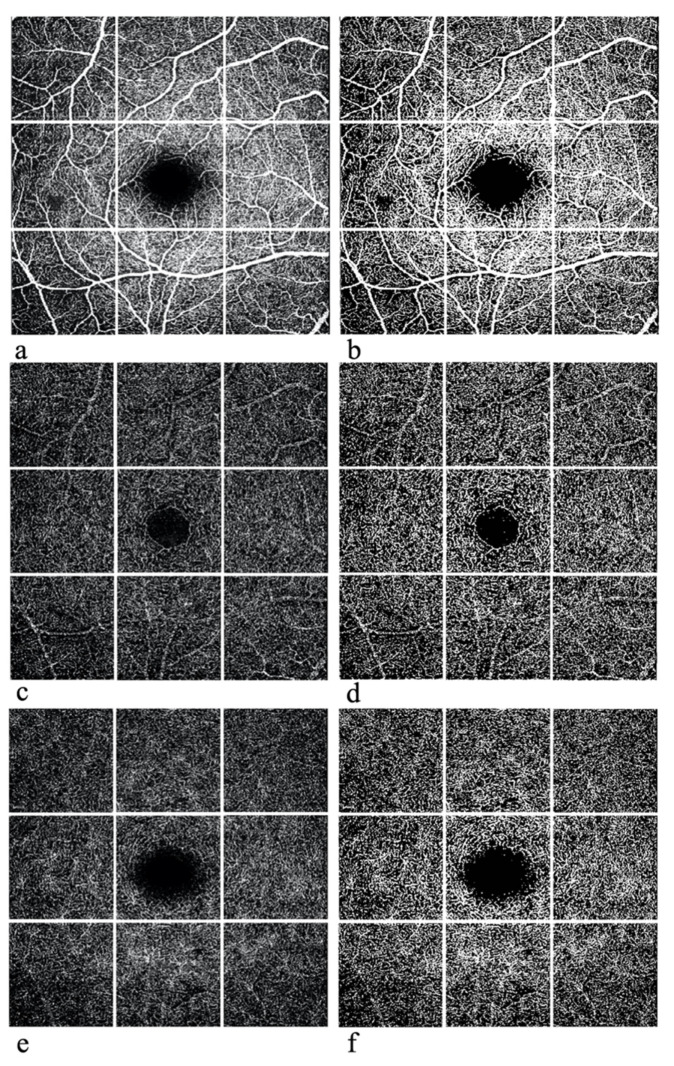
OCT-A scans of the SCP (**a**,**b**), ICP (**c**,**d**), and CP (**e**,**f**) before (**a**,**c**,**e**) and after (**b**,**d**,**f**) processing with Image J.

**Figure 2 diagnostics-12-01908-f002:**
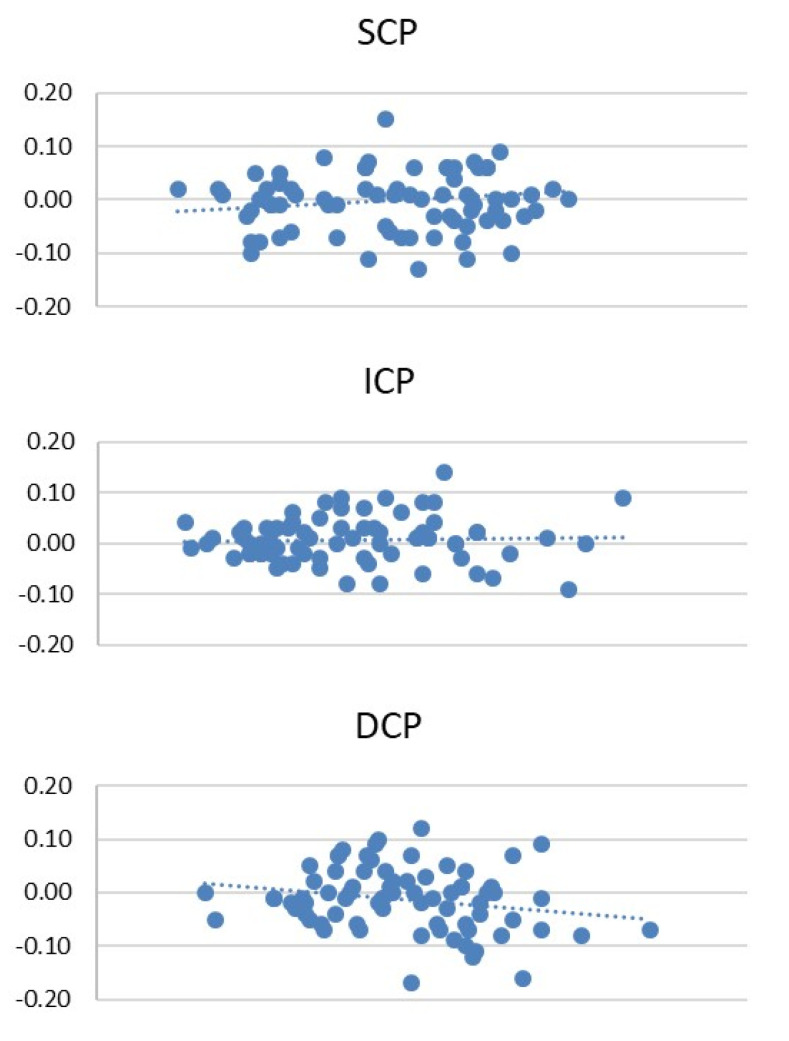
Bland–Altman plots of the first and the second scan of SCP, ICP and DCP.

**Figure 3 diagnostics-12-01908-f003:**
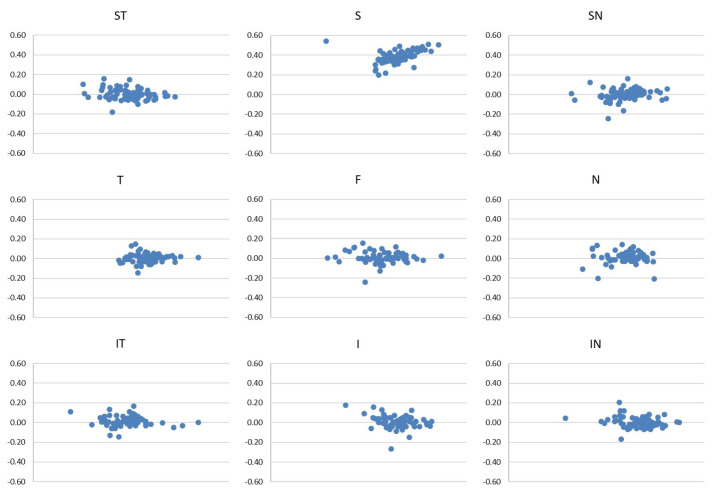
Bland–Altman plots of the first and the second scan of each sector of SCP.

**Figure 4 diagnostics-12-01908-f004:**
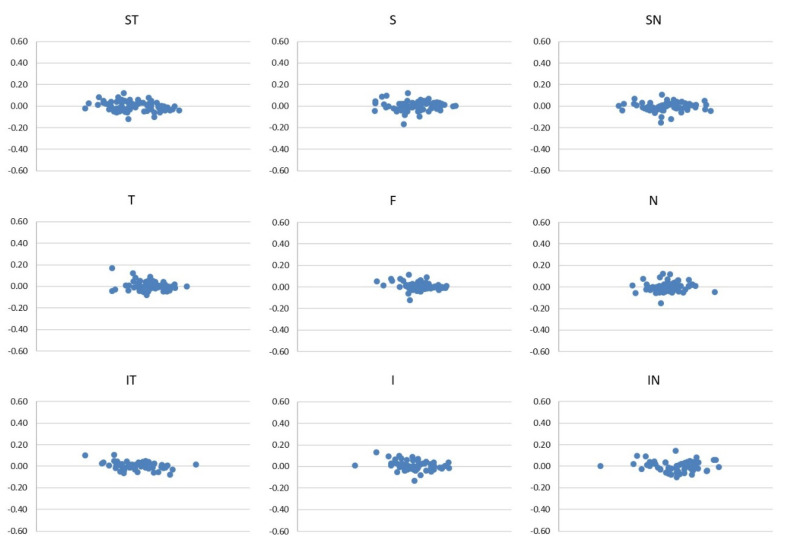
Bland–Altman plots of the first and the second scan of each sector of ICP.

**Figure 5 diagnostics-12-01908-f005:**
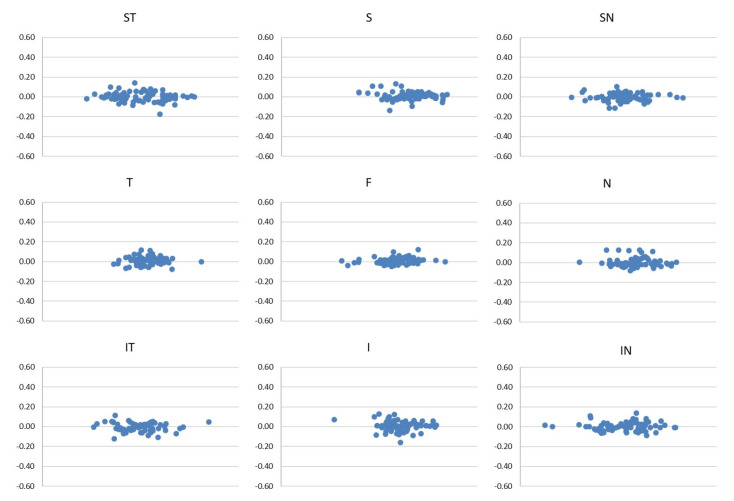
Bland–Altman plots of the first and the second scan of each sector of DCP.

**Table 1 diagnostics-12-01908-t001:** Foveal avascular zone analysis.

		*Mean* *± SD*	*Range (min–max)*	*Coefficient of Variation*	*Coefficient of Repeatability*
***SCP***	*Scan 1*	0.36 ± 0.13	0.11–0.72	0.36	0.12
	*Scan 2*	0.37 ± 0.12	0.09–0.57	0.33	
***ICP***	*Scan 1*	0.23 ± 0.10	0.09–0.53	0.43	0.09
	*Scan 2*	0.23 ± 0.09	0.08–0.48	0.43	
***DCP***	*Scan 1*	0.42 ± 0.11	0.14–0.73	0.27	0.12
	*Scan 2*	0.43 ± 0.12	0.15–0.80	0.29	

*SCP: Superficial capillary plexus; ICP: Intermediate capillary plexus; DCP: deep capillary plexus; SD: standard deviation.*

**Table 2 diagnostics-12-01908-t002:** Overall vessel density analysis.

		*Mean ± SD*	*Range (Min–Max)*
***SCP***	*Scan 1*	36.05 ± 9.01	27.02–45.48
	*Scan 2*	35.33 ± 9.92	26.61–45.22
***ICP***	*Scan 1*	21.87 ± 5.00	18.16–25.55
	*Scan 2*	21.32 ± 5.56	17.96–25.33
***DCP***	*Scan 1*	23.84 ± 6.53	20.32–29.71
	*Scan 2*	23.20 ± 6.84	19.80–29.14

*SCP: Superficial capillary plexus; ICP: Intermediate capillary plexus; DCP: deep capillary plexus; SD: standard deviation.*

**Table 3 diagnostics-12-01908-t003:** SCP vessel density analysis.

*Sector*		*Mean* *± SD*	*Range (Min–Max)*	*Coefficient of Variation*	*Coefficients of Repeatability*
***ST***	*Scan 1*	29.23 ± 5.93	15.09–42.12	0.20	0.11
	*Scan 2*	29.42 ± 7.12	9.95–44.68	0.24	
***S***	*Scan 1*	39.07 ± 5.20	16.98–51.47	0.13	0.13
	*Scan 2*	38.89 ± 6.56	19.92–54.21	0.17	
***SN***	*Scan 1*	35.25 ± 6.92	17.30–51.31	0.20	0.12
	*Scan 1*	35.45 ± 5.90	18.76–50.49	0.17	
***T***	*Scan 1*	35.19 ± 4.95	24.30–51.00	0.14	0.09
	*Scan 2*	34.88 ± 4.83	23.67–50.00	0.14	
***F***	*Scan 1*	34.88 ± 6.70	16.79–53.48	0.19	0.11
	*Scan 2*	34.03 ± 6.90	17.30–51.04	0.20	
***N***	*Scan 1*	48.95 ± 6.84	24.89–61.56	0.14	0.12
	*Scan 2*	46.98 ± 6.92	28.88–69.99	0.15	
***IT***	*Scan 1*	28.63 ± 6.45	16.76–50.57	0.23	0.10
	*Scan 2*	27.65 ± 6.49	15.96–50.51	0.23	
***I***	*Scan 1*	38.60 ± 5.04	23.59–49.89	0.13	0.12
	*Scan 2*	38.16 ± 6.24	14.20–50.77	0.16	
***IN***	*Scan 1*	39.82 ± 5.82	19.69–52.35	0.15	0.11
	*Scan 2*	39.40 ± 6.44	15.13–52.14	0.16	

*SCP: Superficial capillary plexus; ST: Superotemporal; S: superior; SN: superonasal; T: temporal; F: foveal; N: nasal; IT: inferotemporal; I: inferior; IN: inferonasal.*

**Table 4 diagnostics-12-01908-t004:** ICP vessel density analysis.

*Sector*		*Mean* *± SD*	*Range (Min–Max)*	*Coefficient of Variation*	*Coefficients of Repeatability*
***ST***	*Scan 1*	18.02 ± 4.30	7.99–25.70	0.24	0.09
	*Scan 2*	18.29 ± 5.03	7.49–28.60	0.28	
***S***	*Scan 1*	22.45 ± 4.20	11.52–29.70	0.19	0.08
	*Scan 2*	22.11 ± 4.08	11.45–29.44	0.18	
***SN***	*Scan 1*	19.86 ± 4.25	9.85–29.88	0.21	0.08
	*Scan 2*	20.30 ± 4.07	10.53–30.49	0.20	
***T***	*Scan 1*	24.26 ± 3.27	13.88–31.64	0.13	0.08
	*Scan 2*	23.64 ± 3.92	17.55–31.64	0.17	
***F***	*Scan 1*	23.03 ± 2.73	14.75–28.60	0.12	0.07
	*Scan 2*	22.66 ± 3.26	12.17–28.12	0.15	
***N***	*Scan 1*	25.51 ± 4.23	15.22–34.08	0.17	0.09
	*Scan 2*	25.33 ± 3.74	15.91–38.91	0.15	
***IT***	*Scan 1*	19.36 ± 3.40	13.13–30.76	0.18	0.06
	*Scan 2*	18.60 ± 4.19	14.01–28.98	0.23	
***I***	*Scan 1*	22.47 ± 3.38	11.25–29.99	0.15	0.08
	*Scan 2*	21.89 ± 4.26	8.03–28.94	0.20	
***IN***	*Scan 1*	21.85 ± 4.26	17.72–32.34	0.19	0.09
	*Scan 2*	22.15 ± 4.48	17.69–30.19	0.20	

*ICP: Intermediate capillary plexus; ST: Superotemporal; S: superior; SN: superonasal; T: temporal; F: foveal; N: nasal; IT: inferotemporal; I: inferior; IN: inferonasal.*

**Table 5 diagnostics-12-01908-t005:** DCP vessel density analysis.

*Sector*		*Mean* *± SD*	*Range (Min–Max)*	*Coefficient of Variation*	*Coefficients of Repeatability*
***ST***	*Scan 1*	20.46 ± 5.35	8.14–30.90	0.26	0.09
	*Scan 2*	20.34 ± 5.92	8.89–32.73	0.29	
***S***	*Scan 1*	26.19 ± 4.45	15.17–34.98	0.17	0.08
	*Scan 2*	25.37 ± 4.72	13.24–35.60	0.19	
***SN***	*Scan 1*	20.95 ± 4.66	4.58–21.37	0.22	0.08
	*Scan 2*	26.87 ± 4.25	9.48–33.60	0.21	
***T***	*Scan 1*	27.47 ± 4.13	16.82–40.45	0.15	0.07
	*Scan 2*	26.69 ± 4.08	18.78–40.45	0.15	
***F***	*Scan 1*	23.94 ± 4.36	11.59–33.93	0.18	0.06
	*Scan 2*	22.90 ± 3.64	11.89–33.38	0.16	
***N***	*Scan 1*	29.58 ± 5.06	15.52–40.18	0.17	0.09
	*Scan 2*	29.26 ± 5.04	15.02–40.32	0.17	
***IT***	*Scan 1*	19.70 ± 5.22	8.57–43.03	0.27	0.08
	*Scan 2*	20.41 ± 5.31	9.11–43.09	0.26	
***I***	*Scan 1*	24.89 ± 4.09	14.39–33.87	0.16	0.10
	*Scan 2*	24.16 ± 4.49	7.27–32.37	0.19	
***IN***	*Scan 1*	20.99 ± 5.54	5.69–31.43	0.26	0.09
	*Scan 2*	20.65 ± 5.40	4.27–32.06	0.26	

*DCP: Deep capillary plexus; ST: Superotemporal; S: superior; SN: superonasal; T: temporal; F: foveal; N: nasal; IT: inferotemporal; I: inferior; IN: inferonasal.*

## Data Availability

The datasets generated during the current study are available from the corresponding author on reasonable request.

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
