# Peer review of "Reproducibility and Reliability of Spectralis II OCT Angiography Vascular Measurements"

_diagnostics, 2022, doi:10.3390/diagnostics12081908_

Round 1
Reviewer 1 Report
Dear authors,
It was my pleasure to reviewing your manuscript intitle “ Reproducibility and reliability of Spectralis II OCT angiography vascular measurements”.
The manuscript is well-constructed and the topic is relevant, however, some points need to be clarified before the publication.
1) Please add mean the refractive error observed in the study population (results) and the tool used for determinate it (method)
2) Please add the rate of right/left eye included
3) Please clarify if the vessel density analysis and foveal avascular zone in the first and second measurement were obtained by the same physician
4) Please add the author’s role in the study (imaging acquisition, image J elaboration etc)
5) Authors state in methods line 85 “The measurements of VD and FAZ were performed and analyzed 85 by two different experienced examiners. “ please add the statistical adequate test between different examiners
Best regards,
Author Response
Dear reviewer, thanks you for appreciating our study and for your helpful comment.
1) Please add mean the refractive error observed in the study population (results) and the tool used for determinate it (method)
- We edited the text by adding the refractive error observed in the study population and the tool used for determinate it in the results and methods section, respectively (page 2 line 73, and page 4 line 125).
2) Please add the rate of right/left eye included
- We added the value, as required (page 4 line 121)
3) Please clarify if the vessel density analysis and foveal avascular zone in the first and second measurement were obtained by the same physician
- We thank the reviewer for this comment. We edited the text according to the comment. (page 2 line 86).
4) Please add the author’s role in the study (imaging acquisition, image J elaboration etc)
- We modified the methods section with the different Author’s role in the study, as required.
5) Authors state in methods line 85 “The measurements of VD and FAZ were performed and analyzed 85 by two different experienced examiners. “ please add the statistical adequate test between different examiners
- We thank the reviewer for this interesting comment. We edited the text (page 2 line 87) and added the statistical adequate test (page 4 line 116), as suggested.
Reviewer 2 Report
In this study, Pastore et al. evaluated reproducibility and reliability of Heidelberg Spectralis II OCTA vascular measurements. I congratulate the authors for their work. A simple, elegant, and important study. I only have one minor comment:
Post-processing using ImageJ is does not represent clinical practice. However, at this present stage, these measurements are not readily available using native software, so the method employed by the authors is the only available tool. Nevertheless, it should be stressed as a limitation in the discussion, that the results of this study; the good reproducibility and reliability of OCT-A vascular measurement; are based on post-processing steps performed by ImageJ, and not native software. Therefore, if/when the native software introduces possibility of FAZ and vessel density measurements, the findings of this study cannot be generalized to such software and another study will be warranted.
Author Response
Post-processing using ImageJ is does not represent clinical practice. However, at this present stage, these measurements are not readily available using native software, so the method employed by the authors is the only available tool. Nevertheless, it should be stressed as a limitation in the discussion, that the results of this study; the good reproducibility and reliability of OCT-A vascular measurement; are based on post-processing steps performed by ImageJ, and not native software. Therefore, if/when the native software introduces possibility of FAZ and vessel density measurements, the findings of this study cannot be generalized to such software and another study will be warranted.
- We completely agree with the reviewer. We edited the text by adding this point as a limitation of the study at the end of the discussion.